# *Homo cerevisiae*—Leveraging Yeast for Investigating Protein–Protein Interactions and Their Role in Human Disease

**DOI:** 10.3390/ijms24119179

**Published:** 2023-05-24

**Authors:** Florent Laval, Georges Coppin, Jean-Claude Twizere, Marc Vidal

**Affiliations:** 1Center for Cancer Systems Biology (CCSB), Dana-Farber Cancer Institute, Boston, MA 02215, USA; florent_laval@dfci.harvard.edu (F.L.); georges_coppin@dfci.harvard.edu (G.C.); 2Department of Genetics, Blavatnik Institute, Harvard Medical School, Boston, MA 02115, USA; 3Department of Cancer Biology, Dana-Farber Cancer Institute, Boston, MA 02215, USA; 4TERRA Teaching and Research Centre, University of Liège, 5030 Gembloux, Belgium; 5Laboratory of Viral Interactomes, GIGA Institute, University of Liège, 4000 Liège, Belgium; 6Division of Science and Math, New York University Abu Dhabi, Abu Dhabi P.O. Box 129188, United Arab Emirates

**Keywords:** yeast, protein–protein interaction, interactome, edgetics, human variome, personalized medicine

## Abstract

Understanding how genetic variation affects phenotypes represents a major challenge, particularly in the context of human disease. Although numerous disease-associated genes have been identified, the clinical significance of most human variants remains unknown. Despite unparalleled advances in genomics, functional assays often lack sufficient throughput, hindering efficient variant functionalization. There is a critical need for the development of more potent, high-throughput methods for characterizing human genetic variants. Here, we review how yeast helps tackle this challenge, both as a valuable model organism and as an experimental tool for investigating the molecular basis of phenotypic perturbation upon genetic variation. In systems biology, yeast has played a pivotal role as a highly scalable platform which has allowed us to gain extensive genetic and molecular knowledge, including the construction of comprehensive interactome maps at the proteome scale for various organisms. By leveraging interactome networks, one can view biology from a systems perspective, unravel the molecular mechanisms underlying genetic diseases, and identify therapeutic targets. The use of yeast to assess the molecular impacts of genetic variants, including those associated with viral interactions, cancer, and rare and complex diseases, has the potential to bridge the gap between genotype and phenotype, opening the door for precision medicine approaches and therapeutic development.

## 1. The Disease-Causing Human Variome—The Curse of Too Much Knowledge

The OMIM database describes over 6200 genetic disorders and ~5000 genes with disease-causing mutations [1,2,3,4]. HGMD [5,6] and ClinVar [7], which annotate potentially causative variants, cover >220,000 variants in 5000 genes. Soon, nearly all Mendelian disorders will have their associated genes identified and many causative mutations sequenced [8,9]. Genome-wide association (GWA) studies have also exploded [10,11,12,13], with >45,000 studies involving ~5000 traits [14,15,16]. Although GWA studies generally identify tagSNPs, i.e., correlative but not necessarily causative variants, increased cohort sizes and the development of fine-mapping tools will soon enable the identification of functional SNPs [17]. Thus, a comprehensive description of the functionally relevant disease-causing human variome is in sight.

Human inherited disorders are caused by mutations in genes resulting in disruption of normal cellular function. However, a daunting challenge to characterizing genotype-to-phenotype relationships is understanding the molecular functions these variants perturb. Indeed, well-established mechanistic connections that can spur the development of novel therapies are vanishingly rare. For the vast majority of disease-causing variants, we do not understand their impact on molecular and cellular functions.

The functional interpretation of disease-causing human variants is indeed turning out to be more complicated than the classical one-gene/one-function/one-phenotype model: (i) different mutations in a gene can be associated with distinct disorders (allelic heterogeneity), (ii) a disorder can be caused by mutations in any one of several genes (locus heterogeneity), (iii) many gene products mediate different functions (pleiotropy), (iv) not all individuals affected by a given mutation are affected equally (variable expressivity), and (v) only a subset of individuals carrying a mutation are affected by the disease at all (incomplete penetrance) [18,19].

Currently, most variants identified by clinicians cannot be confidently classified and are reported as variants of uncertain significance (VUS). In fact, there are currently 840,000 coding VUSs reported in ClinVar, which is approximately five times the number of annotated pathogenic variants (Figure 1). This highlights the urgent need for reliable methods to assess the clinical significance of genetic variations.

## 2. Edgetics—The Cutting Edge

A system can be defined as an arrangement of essential parts or subsystems that are interconnected and interdependent, operating in consonance with a set of rules to form a unified whole exhibiting behavior or meaning that the individual constituents do not have and is arranged to achieve a specific objective. This definition can be dissected into five distinct characteristics. First, a system implies structure and order between its different components, a feature that is referred to as an organization. Second, a system is defined by its interactions, i.e., how each constituent functions with other parts. Third, a system’s components are interdependent, as one part is dependent on the output of another part or subsystem for proper functioning. Then, the different components are integrated and tied together to allow the system to eventually achieve a central objective. Numbers within mathematical operations are organized in a well-defined order (organization) and interconnected by mathematical symbols (interactions). The outcome of the first operation will define the outcome of the second, which will, in turn, affect the result of the next (interdependence), and the integration of all operations eventually leads to the final result (central objective). Another example of a system consists of the air traffic control system, which consists of a complex network of subsystems, such as radar systems, communication systems, and navigation systems, that are connected (organization) through a network of computer systems (interactions) that provide real-time information about the location and movement of aircraft (integration), and that work together to ensure the safe and efficient movement of aircraft through the airspace (central objective). The system also includes contingency plans for dealing with unexpected events, such as equipment failures, severe weather, and security threats (interdependence). Likewise, the financial system is a complex network of institutions, markets, and individuals that are interconnected (organization) through a network of transactions and financial instruments such as loans and bonds (interactions) that facilitate the flow of money and credit throughout the economy (integration), ensuring that the financial system remains stable and resilient and that it can continue to provide the necessary funding to support economic growth and development (central objective).

Systems biology is an interdisciplinary approach to studying complex biological structures, including cells, tissues, and organisms. It aims to investigate biological systems in a holistic manner by taking into account the interactions between different components of the system, such as genes, proteins, metabolites, and signaling molecules. Moreover, it seeks to uncover the underlying mechanisms that govern the behavior of biological systems, identify the key regulatory pathways and feedback loops that control their function, and understand how these structures work and how they respond to perturbations. Systems biology relies on a wide range of experimental techniques, including but not restricted to genomics, transcriptomics, proteomics, metabolomics, and imaging. One of the primary assets, and challenges, of systems biology lies in the integration of these different types of data together to generate a thorough, global representation of biological systems. These models can ultimately be used to simulate the behavior of the system under varied conditions and to make predictions about how it will respond to perturbations. Systems biology has many applications in fundamental biology and medicine, including drug discovery, personalized medicine, and the development of new therapies for diseases. It hence undoubtedly represents a necessary approach that has the potential to transform our understanding of complex biological systems and to provide new insights into the fundamental principles that govern life.

The one gene-one function model is a concept that proposes that each gene in an organism’s genome is responsible for producing a single functional product, such as a protein, and that this product has a specific and well-defined function in the cell or organism. However, this model is circumscribed and rapidly reaches limitations, as it is not always accurate, particularly in more complex organisms. It often oversimplifies the complexity of genetic interactions and the role of genes in the development and function of organisms. In reality, many genes can have multiple functions depending on the cellular context, the developmental stage of the organism, or the environmental conditions. Additionally, some genes may, for instance, produce non-coding RNAs or other regulatory molecules that can influence the expression or function of other genes. Another drawback of the one gene-one function model is that it does not account for genetic redundancy or compensation. In some cases, multiple genes may be involved in performing a specific function, and the loss of one gene may be compensated for by the function of another gene. This can make it difficult to identify the specific function of each individual gene. Furthermore, many genetic disorders are caused by mutations in genes that affect multiple functions rather than just one function. For example, mutations in the BRCA1 and BRCA2 genes, which are associated with an increased risk of breast or ovarian cancer, affect multiple cellular processes, including DNA repair, transcriptional regulation, and cell cycle control. Overall, while the one gene-one function model can be useful in understanding basic genetic mechanisms, it presents weaknesses in accurately describing the dynamic and systemic nature of how genes function in complex organisms.

Edgetics is the study of the rewiring of interactions in a context that is alternative to the one used to establish a reference interactome map. While interactomics establishes reference interaction networks by mapping interactions, edgetics aims to annotate and understand the structure and function of edges composing these biological networks [20]. The term edgetics is derived from the word edge, which refers to the connections between the different constituents, or the nodes, of a network. The goal of edgetics is to identify key edges (i.e., protein–protein interactions, metabolic reactions, regulatory interactions, etc.) that are critical for the function of the network and to determine how these edges are regulated and coordinated to achieve specific cellular functions. This graph theory has been applied to a wide range of biological systems, from single cells to entire organisms. For example, it has been used to study the regulation of metabolic networks, the signal transduction pathways involved in cancer and other diseases, and the gene regulatory networks that control development and differentiation in multicellular organisms. One of the key advantages of edgetics is that it allows one to survey biological networks in a holistic manner, thoroughly considering their interaction and interdependence facets rather than focusing on individual components in isolation. By taking a network-level perspective, edgetics can reveal emergent properties of the network that are not apparent from the properties of individual components. Genes, and their products, mediate cellular functions by assembling into complex systems linked through biochemical and physical interactions. Indeed, no protein functions in isolation. Therefore, a better understanding of the genotype-to-phenotype relationships, e.g., the impact of a genetic variant on a molecular function, requires determining the global effect of such variants on the whole system rather than on a sole, isolated gene. Edgetics is capable of providing new insights into the complex interactions between genes, proteins, and other cellular components and guiding the development of new therapies. Here are a few examples of how the study of protein–protein interactions (PPIs) can help understand human diseases. In the field of cancer, PPIs are critical for regulating cell growth and division, and alteration of these interactions can contribute to the development and progression of cancer. For example, the interaction between the tumor suppressor protein p53 and the oncoprotein MDM2 is disrupted in many types of cancer, leading to the inactivation of p53 and uncontrolled cell growth [21,22]. Next, neurodegenerative diseases are also clearly impacted by protein–protein interactions that are important for maintaining the structure and function of neuronal cells, and disruption of which can lead to neurodegeneration. For instance, the accumulation of misfolded proteins, such as amyloid plaques in the brain of individuals with Alzheimer’s disease, disrupts PPIs and leads to neuronal cell death [23]. Furthermore, protein–protein interactions are crucial in understanding infectious diseases, since they are critical for the replication and spread of many pathogens, and targeting these interactions can be an effective strategy for developing new antiviral and antibacterial drugs [24]. For example, the interaction between the human immunodeficiency virus (HIV) protein gp120 and the host protein CD4 is essential for viral entry into host cells [25,26], and blocking this interaction can prevent viral replication. The same applies to the interaction between the SARS-CoV-2-encoded NSP16 protein and the host protein USP25, which seems to be hijacked by the virus to protect itself from ubiquitination and degradation by the host defense machinery [27].

In conclusion, the study of protein–protein interactions provides important insights into the molecular mechanisms of diseases and can help identify potential targets for therapeutic intervention.

## 3. Yeast, This Tiny but Mighty Organism

All organisms share biological processes mediated by gene products conserved through evolution [28]. While our eagerness to solve the puzzle these processes represent is mainly anthropocentric, biological functions are more easily unraveled in simpler model organisms [29]. Of particular interest, yeast is a unicellular eukaryotic organism which makes it extremely easy to manipulate in the laboratory. It is characterized by rapid growth and, for instance, allows simple conversion between easily distinguishable haploid and diploid forms. The haploid nature of yeast cells allows for the direct observation of the effect of a genetic alteration (e.g., single nucleotide polymorphism, gene disruption, etc.) on the phenotype since it excludes confounding factors arising from genetic variations such as heterozygosity. Furthermore, many selection markers have been identified and can efficiently be used in yeast. Although its genome contains ~6200 protein-coding genes, which represents 3 to 4 times less than the number of protein-coding genes in the human genome, yeast represents a system comprising most major biological mechanisms. In particular, genetic conservation between humans and yeast is very high [30]. After the publication of the yeast genome in 1996, it has been estimated that about 30% of human disease genes have a functional homolog in yeast [31]. One might however argue that DNA sequence conservation between two organisms does not guarantee the transferable functionality of the gene products from one to the other. For instance, post-translational modifications (PTMs), which refer to covalent processing events of amino acid side chains that change properties of proteins after their synthesis by the ribosomes [32], and disruption of which can lead to aberrant phenotypes and various diseases, are rather divergent between the two organisms. There indeed exist more than 400 different known PTMs in humans [33], and not all are conserved in yeast. Some common PTMs, such as phosphorylation, ubiquitination, acetylation, and glycosylation, occur in both yeast and human proteins, but other more complex PTMs, such as O-GlcNAcylation, that is associated with either the etiology or pathology of neurodegenerative disorders [34], have limited information in yeast and seem to be specialized to human cells. Nonetheless, Kachroo and colleagues empirically attempted to systematically replace a fraction (414) of about 2000 essential yeast genes with their human orthologs and demonstrated that the deletion of about half of them could be complemented by their human counterpart [35,36], showing that such divergence does not completely preclude one from using yeast to interrogate the functionality of human biomolecules.

In view of these attributes, yeast represents an incredibly powerful and popular structure for studying basic biological processes and for investigating the molecular mechanisms underlying many human diseases. It indeed provides a powerful platform for investigating a wide range of biological questions and has provided numerous insights into the workings of the cell and the mechanisms underlying human diseases [37]. In addition to being employed as a model organism, yeast has also proven extremely valuable as a test tube or an experimental environment to produce and survey exogenous biomolecules for specific characteristics, such as interactions between human proteins for example.

On the one hand, yeast (i.e., not restricted to the *S. cerevisiae* species) as a model organism paved the way for understanding basic biological processes such as gene expression, protein synthesis, and cell division. Oftentimes, studies in model organisms reveal the first clues to the identity of a genetic defect in human disease. For example, MSH2 and MLH1, two genes involved in DNA mismatch repair mechanisms and the maintenance of the genome, and deleterious mutations which can lead to Lynch syndrome and non-polyposis colorectal cancer, were identified in yeast prior to being sequenced in humans [38]. Then, numerous physiological pathways have been extensively studied using the eukaryotic cell, including cellular signaling pathways such as stress response [39,40], metabolism, and cell cycle regulation [41,42]. For example, Lee Hartwell used temperature-sensitive mutants, i.e., genetic variants that have a wild-type phenotype at permissive temperature but exhibit an aberrant phenotype at restrictive temperature, and screened thousands of yeast variants for such phenotypes. Through the implementation of such a systematic and unbiased approach, only achievable in the simple model organism, he identified multiple genes that are important for cell division. For instance, temperature-sensitive alleles of CDC28, a yeast gene encoding a cyclin-dependent kinase (CDK) catalytic subunit that is required for cell division, were discovered in these experiments. When cdc28 mutants are grown at restrictive temperatures, the gene product is inactivated, and yeast cells arrest in the cell cycle at specific checkpoints [43]. These screens were conducted in an organism evidently much simpler than humans, which represents a biological structure making such experimental procedures inaccessible, and yet, the reverberation of the discovery of CDKs on our understanding of fundamental human biology is indisputable. In addition, yeast has also proven valuable as a model system for aging research as it undergoes a process called replicative aging, in which the mother cell divides asymmetrically, producing a daughter cell with younger age and a mother cell that gets older with each division [44]. Researchers have been using yeast to study the genetic and molecular factors that contribute to aging, such as the accumulation of oxidative damage [45,46,47] or the dysregulation of protein quality control mechanisms [48,49]. As yeast can be easily manipulated to introduce mutations in mitochondrial genes, it has frequently been used to study mitochondrial functions, including mitochondrial DNA isolation [50] and replication [51,52], respiration [53,54,55], and protein import [56,57]. Yeast also played a pivotal role and still continues to do so in disease modeling, including neurodegenerative diseases, cancer, and infectious diseases. By expressing disease-related genes in yeast and studying their effects on yeast growth and other phenotypes, one can gain important insights into the mechanisms underlying these diseases [58]. For example, the expression of a mutant p53 gene in yeast cells that are deficient in endogenous p53 allows one to assess its functional activity by directly and simply measuring the growth of the yeast cells [59]. Next, genetic interactions, where the effect of one gene is modified by the presence or absence of another gene or mutation, represent one of the most incredible discoveries allowed by the unicellular organism [60]. The rapid identification and characterization of genes that are related to another particular gene of interest provide important insights into the functions of these genes. In this context, Reddy and Desai presented a mathematical model of genetic interactions in complex traits and showed that even in a simplified model, global epistasis could arise due to non-linear interactions between genes [61]. They furthermore showed that global epistasis could be a major factor in the evolution of complex traits and emphasized the fact that understanding the nature of genetic interactions and their effects on complex traits is important for developing effective strategies for genetic engineering and personalized medicine. Still from a perspective of genetic interactions, Jerison et al. investigated the genetic basis of adaptability and pleiotropy using yeast [62]. They employed a large collection of yeast strains and showed that genetic variation could affect the ability of the yeast to adapt to new environments and the extent of pleiotropy of certain genes, which are critical to understanding for predicting the evolutionary trajectories of organisms and for developing effective strategies for genetic engineering and personalized medicine [62]. 

On the other hand, yeast serves as a test tube, or an experimental environment, to efficiently test numerous exogenous biomolecules in a high-throughput and unbiased manner. First, protein folding and quality control are commonly assessed using yeast as a tool. The cells can be easily manipulated to express mutant proteins that have difficulty folding or that are prone to aggregation [63,64,65,66,67,68]. This can help us understand how cells detect and degrade misfolded proteins and how protein misfolding can lead to disease. Another field that largely benefited from using this organism as a tool is synthetic biology, which involves engineering biological systems with novel functions [69]. Yeast can be used as a host for the production of heterologous proteins [70,71], metabolic engineering [72,73], and the construction of synthetic gene networks [74,75,76]. In a more translatable research perspective, yeast can also be used for drug discovery by screening large libraries of compounds for their ability to affect yeast growth or other phenotypes [77,78], and it can be engineered to express human disease-related proteins, allowing for the screening of compounds for their ability to modulate the activity of these proteins [79]. Finally, in the field of biophysical interactions, yeast is frequently and efficiently used to study direct and binary protein–protein interactions as it represents the only experimental structure allowing one to screen and test these biophysical interactions in a robust, reproducible, and high-throughput manner.

## 4. Yeast, or the Room of Requirement for Interactome Mapping

As alluded to above, yeast is considered the easiest eukaryotic organism to study and manipulate, providing significant breakthroughs for researchers studying conserved molecular processes shared with human cells. Additionally, yeast’s ability to express exogenous genes in a cellular context closely aligned with typical physiological human cells makes it a valuable tool for systems biology and studying human diseases. It has therefore been a crucial model organism for interactome mapping and systems biology. Systematic mapping of protein–protein interactions began in model organisms. *Saccharomyces cerevisiae* was the first organism where a systematic mapping of protein–protein interactions was initiated in 1997 [80]. In 2000, Walhout et al. used the yeast two-hybrid (Y2H) system to map binary and direct interactions between *C. elegans* proteins involved in vulval development [81]. Uetz et al. [82] and Ito et al. [83] published the first proteome-scale protein–protein interaction maps of *S. cerevisiae* around the same time. Leveraging the first *C. elegans* ORFeome that was published in 2003 [84], the interactome network of this organism was mapped at a proteome scale in 2004 [85]. The Arabidopsis Interactome Map was then generated in 2008 by the Arabidopsis Interactome Mapping Consortium [86]. In 2016, an interspecies protein–protein interaction network between yeast and human was created, revealing function-interaction relationships through evolution beyond human-yeast homologs [87]. In 2023, a *D. melanogaster* reference interactome map was published, combining the latest PPI mapping efforts with previous datasets [88].

The human binary and direct interactome have also extensively been established by leveraging the power of the Y2H system. In 2005, Rual and colleagues published a study that aimed to identify and map human protein–protein interactions on a large scale with the ultimate goal of gaining insights into the molecular mechanisms underlying complex biological processes [89]. The authors noted that while some interactions have been investigated in detail, there is still much to learn about the larger network of interactions that underlie these processes. For this first effort on such a large scale, the high-throughput Y2H system was used to screen for interactions among a set of 3600 human proteins. A total of 4549 interactions were identified and used to construct a protein–protein interaction network, which was highly interconnected, with many proteins interacting with multiple partners. Several densely connected subnetworks, or modules, which were enriched for proteins with related functions, were identified. For example, the authors described a module involved in RNA splicing, as well as modules involved in cell cycle regulation and signal transduction. They also noted that many of the interactions they detected had not been previously reported, highlighting the importance of large-scale screening approaches to uncover new interactions in an unbiased manner. While one should bear in mind that this network is incomplete, it represents a significant first step toward a deeper understanding of how proteins interact in the human system. In addition to the study by Rual and colleagues, Stelzl et al. also reported in 2005 a large-scale human PPI map using a similar Y2H assay, with the main difference being the use of the LexA DNA-binding domain instead of Gal4′s [90]. These first unbiased maps have highlighted interactions involving disease-associated genes, underscoring the potential for human interactome maps to provide a systems-level understanding of disease development and mechanisms.

Later, in 2014, Rolland and colleagues built on this previous work by Rual et al. and mapped protein–protein interactions by expanding the coverage of the human interactome network and improving the quality of the interactions identified [91]. Back then, this represented the largest experimentally determined binary interaction map, reporting 13,944 PPIs among 4303 proteins. The resulting network was even more connected than the first graph, with many proteins participating in multiple interactions. Several densely connected communities that are enriched for proteins with related functions were again identified. This study, in turn, served as a foundation for further efforts to map the human interactome on a larger scale, using a variety of experimental and computational approaches.

More recently, Luck and colleagues published a third iteration of the effort to map the human interactome [92]. In 2020, they released a high-quality reference map of the human binary protein interactome, which represents the set of pairwise interactions between proteins. Still using the high-throughput Y2H system, they screened interactions among a set of 17,408 human proteins, representing approximately 85% of the protein-coding genes in the human genome [93]. The resulting interactome dataset contains 52,569 high-confidence binary interactions, representing a significant expansion of the previous binary interactome maps. Combined with previous systematic mapping efforts at the Center for Cancer Systems Biology (CCSB), the interaction network comprises 64,006 binary PPIs involving 9094 proteins (http://www.interactome-atlas.org). Thanks to a much higher network density, the identification of densely connected modules that are enriched for proteins with related functions was facilitated. This reference map of the human binary interactome they constructed, in conjunction with other omics data, represents an unprecedented resource for understanding the organization and function of the human proteome, as well as for identifying new drug targets and biomarkers. 

In addition to systematic mapping of protein interactions in model organisms, systems biology has enabled extensive investigation of human diseases using the two-hybrid system in yeast. Using this approach, virus PPI networks and virus-host PPI networks have been generated. For example, Calderwood et al. used this system to study herpes viruses, revealing a reference interactome between the Epstein-Barr virus and human proteins [94] as well as potential functions of uncharacterized virus proteins. More recently, Vandermeulen et al. systematically analyzed the interactome of key effectors of the oncoviral proteins Tax and HBZ in HTLV-1 [95]. In 2012, Rozenblatt-Rosen et al. systematically studied host interactome and transcriptome network perturbations caused by DNA tumor virus proteins, resulting in integrated viral perturbation data reflecting rewiring of host cell networks and highlighting pathways that go awry in cancer, such as Notch signaling and apoptosis [96]. Network modeling has also linked breast cancer susceptibility and centrosome dysfunction [97]. Mendelian diseases have been studied in the spectrum of protein–protein interactions, with a study on the two ataxia disease-causing genes ATXN7 and CACNA1A that identified new protein partners that may explain comorbidity of ataxia with other genetic diseases such as macular degeneration [98]. Interactome maps can also be used to study complex diseases, such as autism spectrum disorders (ASD). For instance, protein interaction networks revealed high connectivity between SHANK and TSC1, two ASD-related proteins, suggesting shared common molecular pathways [99]. The Autism Spliceform Interaction Network, a protein interaction network of brain-specific alternatively spliced isoforms of genes related to ASD, revealed that about half of protein interactions were isoform-specific [100].

In summary, the two-hybrid system in yeast has been widely used for mapping biomolecular interactions. According to the IntAct database, which records biophysical interactions from the literature across various organisms, over two-thirds of reported interactions have been identified using this assay (Figure 2A). However, the number of publications reporting two-hybrid interactions is about five times smaller than those reporting interactions using other detection methods (Figure 2B). This further highlights the extent to which the two-hybrid system proves useful for high-throughput interaction mapping.

## 5. The Yeast Assayome—A Versatile Toolbox

Multiple technologies are generally necessary to comprehensively map and characterize any biological system, whether it consists of DNA, RNA, proteins, etc. As illustrated by Choi et al., no single assay is indeed able to identify all biomolecular interactions within a given system [101]. Specifically for protein–protein interactions, using a set of well-defined literature interactions, Choi et al. have shown that individual binary interaction mapping assays have a sensitivity of around 20–30%. Combining multiple assays to map PPIs has several advantages in terms of sensitivity and detection rate. First, different assays can detect interactions that occur under different conditions. For example, some interactions may only occur in the presence of a specific cellular signaling pathway or in response to a certain stressor. By integrating multiple assays, one can increase the chances of detecting these interactions. Second, different assays have different sensitivities for detecting interactions. Some methods are highly sensitive and can detect more transient, less stable interactions, while other protocols can provide information on the specificity of interactions. Third, by using multiple assays, one can reduce the likelihood of false positives or false negatives. Adopting diverse experimental strategies in parallel allows one to increase the confidence in their datasets by orthogonally confirming interactions detected by one assay with another. Finally, implementing several methods side-by-side can provide complementary information about PPIs.

Pertaining to protein–protein interactions, two main classes of assays can be considered: co-complex association or binary and direct assays [102]. 

Co-complex association methods seek the identification of proteins that are concomitantly recruited and come together in a cellular context without determining the direct points of contact between them. These methods are usually directly conducted on cells from the species that are studied (i.e., human cells for mapping the human interactome, yeast cells for the yeast interactome, etc.). The most widely used co-complex association method, the affinity purification-mass spectrometry (AP-MS) strategy, consists in fusing bait proteins to peptide tags that are used to pull them down along with other biomolecules that physically associate with the baits. Identification of these molecules follows by mass spectrometry. Co-fractionation represents another extensively used PPI-mapping by association approach that has the benefit of not introducing exogenously expressed proteins. It instead relies on biochemical separation protocols of cell extracts generating multiple fractions that are then analyzed by mass spectrometry. Protein associations are then inferred by co-elution profiles. These two techniques, while efficient in picking out proteins that associate within cells, lack some granularity in the sense that they do not reveal which molecular species are in direct contact in a protein complex. 

Other protocols are more suitable to address that question, which are referred to as binary and direct methods. These can be conducted in vitro (vN2H), in yeasto (yN2H, Y2H) or in cellulo (mN2H, GPCA, MAPPIT, LuTHy-BRET, LuTHy-LuC, KISS, LUMIER, DULIP). While most of these assays are conducted in mammalian cells, yeast represents a suitable environment for a couple of these techniques: yeast two-hybrid and nanoluc two-hybrid. First, the Y2H system is based on the reconstitution of a transcription factor (TF) summarized as DB-X:AD-Y, where DB is a sequence-specific DNA-binding domain, AD is a transcriptional activation domain, and X and Y are proteins, or protein variants, being tested for interaction [103,104]. Several markers can be expressed from a promoter containing DB-binding sites, whose activation is induced by the DB-X:AD-Y interaction. The DB and AD moieties can be derived from various transcription factors. Two main, distinct yet complementary systems were devised and are predominantly implemented today. On the one hand, the *E. coli* repressor LexA, a well-characterized TF known to bind to the promoter of SOS-response regulatory genes involved in DNA repair mechanisms in the bacterium, is exploited [105]. In this version of the assay, the reporter gene comprises LexA operators fused upstream of a selective marker. On the other hand, the second system takes advantage of the endogenous yeast Gal4 transcription factor [103], which is involved in the activation of genes specialized in the usage of galactose as a source of carbon, such as *GAL1*, *GAL2*, *GAL7,* or *GAL10* [106]. Promoter regions of these Gal4-responsive genes all contain binding sites recruiting the transcription factor. Nevertheless, their architecture varies in terms of the number, sequence, and spacing of TF-binding sites [107,108,109]. This, in turn, allows one to vary the genetic organization of promoters to reach a suitable level of selection stringency. Additionally, artificial promoters have also been created, such as the *SPALn* promoters, which contain the upstream repressing sequence of the yeast *SPO13* promoter and *n* Gal4-binding sites [110]. 

Regardless of the origin of the DNA-binding and activation domains, a variety of options exist with respect to the choice of a reporter gene. The most widely used versions of the yeast two-hybrid system rely on the expression of auxotrophic markers usually involved in biosynthesis pathways of nucleotides or amino acids, thus calling for a growth readout on appropriate dropout media [111]. While a growth readout is a powerful approach to robustly study protein–protein interactions and allows one to perform interaction screens in a high-throughput fashion, the downside is that it often leads to a binary call on the occurrence of a biomolecular interaction and lacks granularity in terms of quantitation. To tackle this challenge, other readout approaches have been devised. An alternative strategy that has been present since the early days of the Y2H system consist in using the LacZ bacterial gene [103], which codes for a beta-galactosidase enzyme hydrolyzing the substrate X-gal, itself colorless, into 5-bromo-4-chloro-indoxyl, which in turn dimerizes to produce 5,5′-dibromo-4,4′-dicholoro-indigo, an insoluble pigment leading to the formation of blue colonies. More recently, Heinz and colleagues generated yeast strains harboring a fusion of the *GAL2* gene promoter to the NanoLuc reporter gene [112]. The latter codes for the nanoluciferase enzyme that relies on the conversion of the substrate furimazine into furimamide, a reaction resulting in the emission of high-intensity, glow-type luminescence. They demonstrated that this bioluminescent version of the Y2H system that uses a continuous readout measure could discriminate light emitted from actual protein–protein interactions from background signals. Yachie et al. incorporated the famous Cre recombination system, originally stemming from the P1 bacteriophage, into the Y2H assay [113]. DNA barcodes are attributed to expression vectors harboring the genes coding for proteins tested for interaction and paired to the latter. Post-mating, selective pressure is applied to yeast cells, and chimeric protein–pair barcodes can be quantified via next-generation sequencing, thus allowing the identification of the interacting partners. While this strategy originally uses a growth readout, this Barcode Fusion Genetics Y2H was recently taken to the next step by utilizing GFP as a reporter gene and sorting and selecting fluorescent cells via flow cytometry, ultimately allowing one not to rely on a growth phenotype (unpublished). 

In addition to the yeast two-hybrid system, protein complementation assays have proven valuable in studying binary and direct protein–protein interactions. In contrast to Y2H, which relies on the activation of a full-length reporter gene, a complementation assay typically consists in splitting the reporter gene into two halves that are each expressed as recombinant proteins fused to the interrogated X and Y proteins. In the instance where X and Y interact, the two fragments of the reporter protein are brought in close proximity, which allows the reconstitution of the full-length (enzymatically) active species. While most of these systems, such as GPCA or YFP-PCA, are carried out in a mammalian cell context, the NanoLuc two-hybrid system (N2H) devised by Choi et al. can be conducted in vitro, in yeasto and in cellulo, using a unique set of expression vectors. The strength of the system is based on the engineering of a tripartite promoter allowing expression of the queried genes in these three distinct environments. 

Both Y2H and N2H technologies have the advantage of being versatile and can be executed in a variety of flavors. First, either method uses a set of two plasmids expressing recombinant proteins, and the combination of fragmented peptide-queried proteins can be swapped. Then, from an architectural perspective, the queried proteins and peptide fragments can be expressed as N-terminal or C-terminal fusions, likely resulting in diverse molecular conformations and hence decreasing the likelihood of potential spatial hindrance. Finally, different levels of gene expression can be achieved by using various origins of replication on the DNA vector backbones.

Regardless of the primary screening method that is chosen for interactome mapping and edgetic functionalization, it is important to validate protein–protein interaction datasets obtained by high-throughput studies to ensure their quality using orthogonal methods, along with benchmark datasets. Orthogonal validation refers to the process of validating results obtained through a given method by using a different, independent approach that provides complementary information. In the context of protein–protein interaction studies, this means that results obtained using a particular high-throughput assay, such as yeast two-hybrid or co-immunoprecipitation, are validated using a different assay or method that provides independent confirmation of the authenticity of these interactions. With respect to the benchmark datasets, they typically consist of well-characterized interaction pairs that serve as a positive reference set (PRS) and a set of protein pairs selected at random that is used as a random reference set (RRS) [114]. Any high-throughput interaction assay can then be optimized to maximize the recovery of the PRS while minimizing the recovery of the RRS interactions. To assess the quality of large-scale datasets obtained using any assay, the recovery rate of a representative sample of interactions can be compared against that of the positive and random set of pairs [115]. Implementation of such an approach helps minimize false positives and false negatives and provides greater confidence in interactome datasets.

## 6. Edgotyping of Genetic Variants—Action in the Interaction

The concept of “edgetic perturbation” in protein–protein interaction networks were extensively discussed by Zhong et al., in 2009, in an article in which the authors highlight the potential of such an approach for understanding the genetic basis of human inherited disorders [20]. In this approach, they focus on perturbations at the protein–protein interaction level rather than the gene level. As a proof-of-concept, the authors analyzed the genetic basis of several inherited disorders, including cystic fibrosis, sickle cell anemia, and Tay-Sachs disease. They constructed edgetic perturbation models of these disorders by identifying the key PPIs disrupted by disease-causing mutations. The work found that the edgetic perturbation models provided new insights into the molecular mechanisms underlying the inherited disorders. For example, they showed that the cystic fibrosis transmembrane conductance regulator (CFTR) protein interacts with a number of other proteins involved in ion transport and cellular signaling and that these interactions are critical for the proper localization and activity of CFTR in the cell membrane. Mutations in CFTR that disrupt these interactions lead to the development of cystic fibrosis [116,117,118,119,120]. Additionally, they showed that the sickle cell anemia mutation disrupts interactions between the hemoglobin protein and other proteins involved in oxygen transport, leading to the characteristic sickle-shaped red blood cells [121,122,123].

In 2013, Sahni and colleagues implemented the concept of “edgotype” as a way to bridge the gap between genotype and phenotype, which is critical for developing personalized medicine [124]. In their article, the authors analyzed the edgotypes of several disease-associated proteins, including the tumor suppressor protein p53 and the Alzheimer’s disease-associated protein amyloid beta. Edgotypes of disease-associated variants were compared with those of non-disease-associated variants to identify specific protein–protein interactions that may be linked to disease. Disease-associated variants often disrupt specific protein–protein interactions that are important for cellular function. For example, the p53 variants associated with cancer disrupted interactions with proteins involved in DNA repair and cell cycle regulation, while the Alzheimer’s disease-associated amyloid beta variants disrupted interactions with proteins involved in synaptic function. This is the demonstration that edgotyping, or the study of edgetic profiles, provides a fundamental link between genotype and phenotype by focusing on the protein–protein interactions that are disrupted by disease-associated variants. 

Two years later, the same group investigated the molecular basis of a larger set of genetic disorders by analyzing the impact of a range of mutations on PPI networks and then identifying disease-associated genes and the network modules they participate in [125]. They hypothesized that understanding the patterns of macromolecular interactions disrupted by disease-associated genes could reveal new insights into the pathogenesis of genetic disorders. Their analysis revealed that disease-associated genes tend to be more connected within the PPI network than non-disease genes, an observation that is consistent with previous studies suggesting that disease genes are more likely to interact with other genes in the same molecular pathway or network. Additionally, these disease-associated genes tend to cluster in specific network modules, suggesting that perturbations in these modules may lead to disease phenotypes. To further explore the functional and structural properties of disease-associated network modules, the authors conducted enrichment analyses to identify biological processes and pathways that were overrepresented within these modules. They found that they tend to be involved in essential cellular processes, such as DNA repair, cell cycle regulation, and protein folding. This observation suggests that perturbations in these modules could have widespread effects on cellular function. The authors next investigated the functional and structural properties of disease-associated network modules and found that they display increased interconnectivity, suggesting a high degree of functional coordination. Overall, Sahni et al. provide insights into the molecular basis of genetic disorders and highlights the importance of considering the network context of disease-associated genes.

Many Mendelian diseases display tissue-specific phenotypes. Yet, disease genes are often uniformly expressed across tissue types [126]. In 2020, Luck et al. showed that perturbation of interactions between uniformly expressed disease-associated proteins and tissue-preferentially expressed proteins could underlie tissue specificity of disease manifestation. Indeed, using pathogenic variants in ten causal proteins tested, seven showed perturbation of PPIs to preferentially expressed interaction partners in the corresponding ‘disease tissues’ [92].

Protein–protein interaction networks represent only a specific set of networks that provide a foundational framework upon which additional layers of functional connections can be built to fine-tune the representation of biological reality. A full understanding of the internal organization of a cell requires the integration of other types of interactome networks, such as transcriptional profiling networks, phenotypic profiling networks, genetic interaction networks, gene regulatory networks, or metabolic networks. While cellular networks based on functional links differ significantly from protein–protein interaction maps in terms of the nature of their connections, they can complement each other and provide valuable insights into biological reality. This creates a reciprocal relationship between the two approaches, which eventually enables the construction of multidimensional maps that follow a series of basic organizing principles and that can be used to better understand genotype-to-phenotype relationships. Vidal et al. exhaustively reviewed why considering perturbations of biological networks is critical in interpreting genetic variation and can shed light on how it relates to phenotypic differences [127].

The edgetic perturbation approach has proven useful in providing a valuable tool for understanding the molecular basis of human inherited disorders. The approach could be used to identify potential drug targets and to develop more personalized therapies based on the specific perturbations present in each patient’s disease. 

## 7. Development of Therapies Targeting PPIs—The Usefulness of Useless Knowledge

The development of therapies using knowledge of protein–protein interactions is an active area of research in the field of drug discovery. In recent years, there has been an increasing focus on targeting protein–protein interactions to develop new treatments for a wide range of diseases, including cancer, infectious diseases, and neurological disorders. One approach to developing therapies based on protein–protein interactions is to identify small molecules or peptides that can disrupt or modulate the interaction. These molecules can be designed to target specific binding interfaces or to stabilize or destabilize the interaction [128,129,130]. Several strategies have been used to identify these molecules, including high-throughput experimental screening, virtual screening, and fragment-based drug discovery. Another approach is to develop therapeutic antibodies that target specific protein–protein interactions. Antibodies can be designed to bind to specific regions of a protein, blocking or disrupting its interaction. Several therapeutic antibodies targeting protein–protein interactions have been approved for use in the clinic, including pembrolizumab directed against the well-known PD-1 protein for the treatment of cancer [131] and adalimumab that specifically binds to the tumor necrosis factor (TNF)-α and used for the treatment of autoimmune diseases [132,133]. In addition to small molecules and antibodies, other strategies for targeting protein–protein interactions include RNA interference (RNAi) and gene editing. RNAi can be used to silence the expression of one of the proteins involved in the interaction, while gene editing can be used to disrupt the gene encoding one of the proteins. In any case, the development of therapies based on protein–protein interactions requires a detailed understanding of the structure and function of the proteins involved in the interaction. This can be achieved through a combination of experimental techniques, including X-ray crystallography, nuclear magnetic resonance spectroscopy, and computational modeling. Overall, the development of therapies based on protein–protein interactions has the potential to provide new treatments for a wide range of diseases. However, it is a challenging area of drug discovery that requires a deep understanding of protein structure and function, as well as expertise in drug design and development.

One example of a drug that has been developed based on the knowledge and study of protein–protein interactions is Venetoclax, which is used for the treatment of chronic lymphocytic leukemia (CLL) and some types of non-Hodgkin’s lymphoma. It is a BH3 mimetic that targets a protein–protein interaction between B-cell lymphoma 2 (BCL-2) and BCL-2-associated X protein (BAX), which are both members of the BCL-2 family of proteins that regulate apoptosis, or programmed cell death. BCL-2 is an anti-apoptotic protein that inhibits cell death, while BAX is a pro-apoptotic protein that promotes cell death. The interaction between BCL-2 and BAX is critical for regulating apoptosis in healthy cells, but it can also be exploited by cancer cells to promote their survival and proliferation. Venetoclax is a small molecule inhibitor that binds to the BH3-binding groove of BCL-2, preventing its interaction with BAX and other pro-apoptotic proteins. This leads to the activation of the apoptotic pathway and the death of cancer cells [134,135]. The drug was developed through a collaboration between the pharmaceutical companies AbbVie and Genentech, and it was approved by the FDA in 2016 for the treatment of CLL with a specific genetic abnormality. 

Another example of a drug developed based on the knowledge and study of protein–protein interactions is Enbrel, which is used for the treatment of autoimmune diseases such as rheumatoid arthritis, psoriasis, and ankylosing spondylitis. Enbrel is a biologic drug that works by inhibiting the activity of the TNF-α protein, which is a key regulator of inflammation and immune response. In autoimmune diseases, the immune system mistakenly attacks healthy tissues, leading to chronic inflammation and tissue damage. TNF-α is a cytokine that plays a major role in this inflammatory response, and inhibiting its activity can help reduce inflammation and relieve symptoms. Enbrel is a fusion protein that consists of a human TNF receptor and the Fc region of a human antibody. The TNF receptor binds to TNF-α and prevents it from interacting with its cell surface receptors, thereby inhibiting its activity. The Fc region of the antibody allows the drug to have a longer half-life and enhances its ability to engage with the immune system [136]. The drug was developed by the pharmaceutical company Amgen and was approved by the FDA in 1998 for the treatment of rheumatoid arthritis. Since then, it has been approved for the treatment of several other autoimmune diseases. 

One last example of a drug developed based on the knowledge and study of protein–protein interactions is Herceptin, which is used for the treatment of certain types of breast cancer. Herceptin targets a protein called human epidermal growth factor receptor 2 (HER2), which is overexpressed in about 20% of breast cancers. HER2 is a receptor tyrosine kinase that promotes cell growth and survival, and its overexpression leads to increased cell proliferation and tumor growth. Herceptin is a monoclonal antibody that binds to the extracellular domain of HER2 and prevents its dimerization, and, thus, activation, thereby inhibiting cell growth and inducing cell death. It also triggers the immune system to attack the cancer cells [137]. The drug was developed by the biotechnology company Genentech, and it was approved by the FDA in 1998 for the treatment of HER2-positive metastatic breast cancer. Since then, it has been approved for the treatment of early-stage HER2-positive breast cancer and metastatic HER2-positive gastric cancer. 

In summary, the development of Venetoclax, Enbrel, and Herceptin are great examples of how the study of protein–protein interactions can lead to the identification of new drug targets and the development of effective therapies for cancer and other diseases.

## 8. Conclusions and Outlook

In the last two decades, extensive sequencing efforts have led to the identification of a vast amount of genetic variation. However, the full potential of genomic data can only be unleashed in concert with high-throughput functionalization technologies. Accurately determining the clinical significance of genetic variants is a pressing need for developing prophylactic measures and therapeutics and currently represents a major bottleneck for delivering on the promise of personalized medicine. 

In this review, we have highlighted the key role of yeast in addressing this urgency. As a eukaryotic organism evidently much simpler than humans, it allows the implementation of experimental procedures that are utterly inaccessible in higher-order eukaryotes, with yet an undeniable repercussion on our understanding of fundamental human biology. The genetic tractability and rapid growth of yeast make it an ideal system for large-scale studies. It has notably played a central role in establishing reference protein–protein interaction maps and the investigation of the potential involvement of these biophysical interactions in human diseases. Leveraging yeast’s strengths as a model system and an experimental tool represents an inevitable step to shed light on the molecular basis of human disease, including cancer, neurodegenerative disorders, and infectious diseases, and bridging the gap between genotype and phenotype.

Overall, yeast has been and will continue to be used extensively as an experimental structure for studying protein–protein interactions and functionalizing mutations involved in human disease. Indeed, while the ideal scenario would consist in relying on the interrogation of rapid, streamlined, and resource-effective computational tools to determine the impact of genetic variation on phenotype, such approaches still leave too much room for uncertainty in the context of precision medicine. The development of prediction algorithms requires extensive training on data generated through empirical, experimental testing, but, despite the seemingly large amount of existing data, computational variant effect predictors still lack accuracy. It appears clear that yeast, because of the significant advantages it offers, will play an essential role in the endeavor of building accurate, decisive models for personalized medicine for years to come.

## Figures and Tables

**Figure 1 ijms-24-09179-f001:**
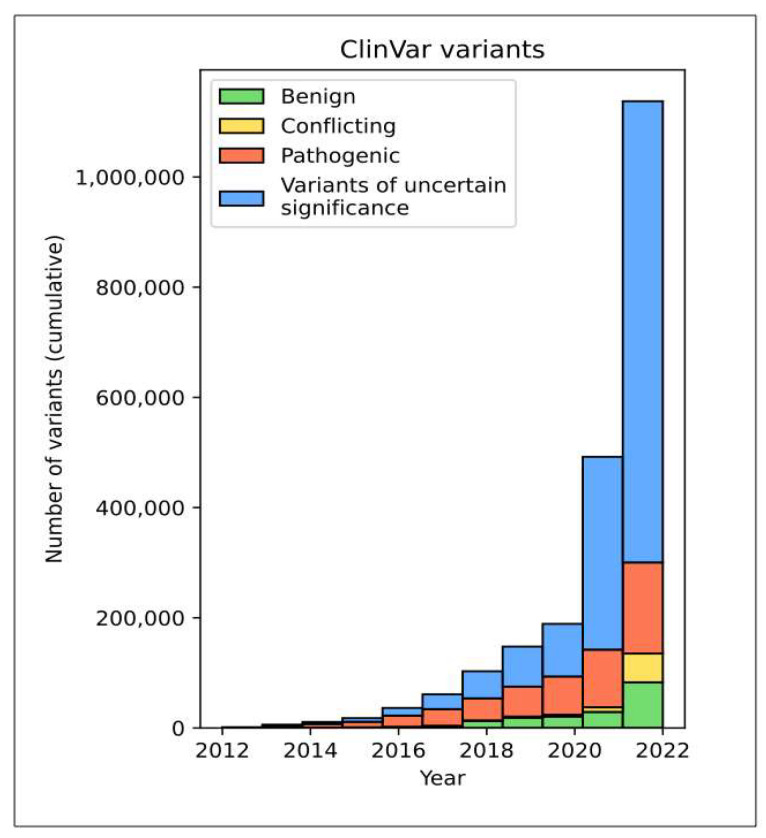
ClinVar number of variants (cumulative) over time.

**Figure 2 ijms-24-09179-f002:**
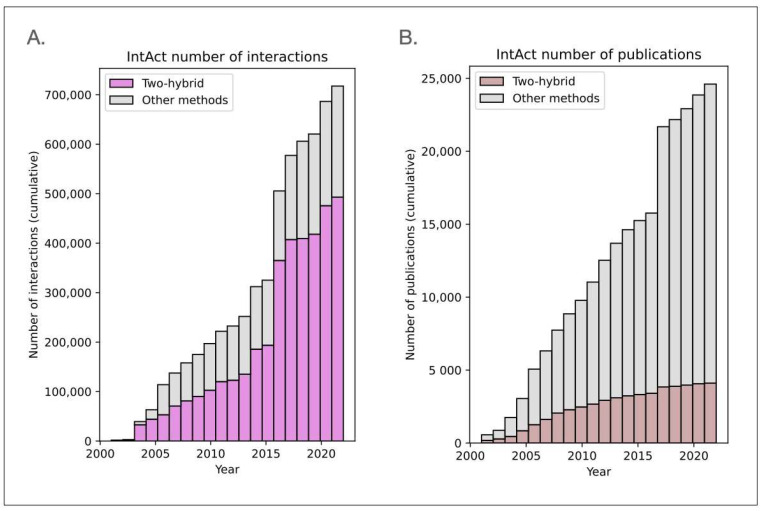
(**A**). IntAct cumulative number of interactions over time; (**B**). IntAct cumulative number of publications (pubmed id) over time (interactions annotated with interaction type as psi-mi:MI:0914 (association), psi-mi:MI:2364 (proximity), psi-mi:MI:0403 (colocalization) were not considered).

## Data Availability

Not applicable.

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
