# Peer review of "Homo cerevisiae—Leveraging Yeast for Investigating Protein–Protein Interactions and Their Role in Human Disease"

_ijms, 2023, doi:10.3390/ijms24119179_

Round 1

Reviewer 1 Report

ijms-2403276

Homo cerevisiae – Leveraging yeast for investigating protein-protein interactions and their role in human disease

Florent Lava, Georges Coppin, Jean-Claude Twizere and Marc Vidal

The authors prepared a review to show yeast is a valuable model organism and as an experimental platform for investigating the molecular basis of phenotypic perturbation upon genetic variation.

This is a quite fascinating review from an interesting point of view. I have only a few suggestions that may help to improve this review.

1) It would be wonderful if the authors could state the protein modifications that are specific to human, not found in yeast. The protein modifications are involved in human diseases in some cases.

2) It would be helpful if the authors could mention that yeast can be at haploid status, which may contribute to show phenotypes.

Author Response

Please, find word document attached.

Reviewer 2 Report

The review article by Laval et al provides a detailed review of the role yeast has played in the development of systems biology approaches in the characterization of human genetic variants. The age of high throughput studies has provided us with innumerable disease-causing human variants for which the impact on molecular and cellular functions is unknown. This review aims to highlight this gap in the knowledge generated by omics studies and discusses how edgetics using yeast has the potential to systematically assess the clinical significance of uncharacterized human genetic variations. The review is a comprehensive report of how yeast has served as a lucrative model for the development of co-complex association and direct methods to identify protein-protein interactions (PPIs) and also facilitated PPI mapping in different model systems, including for human proteome.   The topic is broadly relevant to the field of systems biology.  Overall, the review is very well organized. The referencing is extensive and appropriate. The review will benefit if the authors can also include a brief discussion of how mapping and characterizing the conserved cellular pathways and PPIs in yeast will directly translate into individual-level PPI mapping for developing precision and personalized medicines in humans.

Quality of English language is excellent.

Author Response

Please, find word document attached.
